Does our legal minimum drinking age modulate risk of first heavy drinking episode soon after drinking onset? Epidemiological evidence for the United States, 2006–2014

Cheng Hui G.
Anthony James C. janthony@msu.edu
Department of Epidemiology & Biostatistics, Michigan State University , East Lansing, MI , United States
Patton Bob
Electronic publication date: 2016 Jun 23
Publication date: 2016
Volume: 4
Electronic Location ID: e2153
Received 2016 Apr 23; Accepted 2016 May 30
Copyright: ©2016 Cheng and Anthony
Copyright year: 2016
Copyright holder: Cheng and Anthony
License: This is an open access article distributed under the terms of the Creative Commons Attribution License, which permits unrestricted use, distribution, reproduction and adaptation in any medium and for any purpose provided that it is properly attributed. For attribution, the original author(s), title, publication source (PeerJ) and either DOI or URL of the article must be cited.
License URL: https://creativecommons.org/licenses/by/4.0/

Keywords: Heavy episodic drinking, Newly incident drinkers, Adolescents, United States

Funding: National Institute on Drug Abuse grants K05DA015799 T32DA021129 Michigan State University The study is supported by funds from the National Institute on Drug Abuse grants K05DA015799 (to JCA) and T32DA021129 (to HGC), as well as Michigan State University. The funders had no role in study design, data collection and analysis, decision to publish, or preparation of the manuscript.

==============================
Background. State-level ‘age 21’ drinking laws conform generally with the United States National Minimum Drinking Age Act of 1984 (US), and are thought to protect young people from adverse drinking experiences such as heavy episodic drinking (HED, sometimes called ‘binge drinking’). We shed light on this hypothesis while estimating the age-specific risk of transitioning from 1st full drink to 1st HED among 12-to-23-year-old newly incident drinkers, with challenge to a “gender gap” hypothesis and male excess described in HED prevalence reports.

Methods. The study population consisted of non-institutionalized civilians in the United States, with nine independently drawn nationally representative samples of more than 40,000 12-to-23-year-olds (2006–2014). Standardized audio computer-assisted self-interviews identified 43,000 newly incident drinkers (all with 1st HED evaluated within 12 months of drinking onset). Estimated age-specific HED risk soon after first full drink is evaluated for males and females.

Results. Among 12-to-23-year-old newly incident drinkers, an estimated 20–30% of females and 35–45% of males experienced their 1st HED within 12 months after drinking onset. Before mid-adolescence, there is no male excess in such HED risk. Those who postponed drinking to age 21 are not spared (27% for ‘postponer’ females; 95% CI [24–30]; 42% for ‘postponer’ males; 95% CI [38–45]). An estimated 10–18% females and 10–28% males experienced their 1st HED in the same month of their 1st drink; peak HED risk estimates are 18% for ‘postponer’ females (95% CI [15–21]) and 28% for ‘postponer’ males (95% CI [24–31]).

Conclusions. In the US, one in three young new drinkers transition into HED within 12 months after first drink. Those who postpone the 1st full drink until age 21 are not protected. Furthermore, ‘postponers’ have substantial risk for very rapid transition to HED. A male excess in this transition to HED is not observed until after age 14.

Introduction

Evaluated in the United States (US) and elsewhere in the world, adolescent and young adult drinkers are at increased risk of car crashes, drowning, and a broad range of other adverse events, as compared to same-cohort peers who do not drink. These drink-related adversities now help account for substantial alcohol-attributable disease burdens and social costs (Rehm et al., 2010; Spear & Swartzwelder, 2014). There are additional risks for adolescents who start drinking before late adolescence, and for those who make a fairly rapid transition into heavy episodic drinking (HED), usually defined as five or more drinks in one occasion (Crego et al., 2009; Crews, He & Hodge, 2007).

Consistent with a federal initiative intended to reduce HED and other harms from underage drinking, in most states of the US, the legal minimum age for purchase and autonomous consumption of alcoholic beverages has been set at age 21 years. These ‘age 21’ laws are based on guidelines in the US National Minimum Drinking Age Act of 1984 (NMDAA).

At present, roughly three decades after NMDAA enactment, public health effects of the age 21 threshold continue to be debated (Dejong & Blanchette, 2014; Pitts, Johnson & Eidson, 2014; Wechsler & Nelson, 2010). Some US college presidents, among others, believe that the ‘age 21’ law in the US deserves reconsideration, and express concerns about unhealthy college campus impact of the ‘age 21’ laws, including what they observe about underage HED among college students (Rutledge, Park & Sher, 2008). Nonetheless, a general societal expectation is optimistic, with benefits of the ‘age 21 minimum’ anticipated in the form of lower rates of unhealthy drinking behaviors, economic cost reductions, and improved public safety (Dejong & Blanchette, 2014; Centers for Disease Control and Prevention, 2014; Wechsler & Nelson, 2010).

There may be an evidence-based rationale for setting a minimum legal minimum age for drinking alcohol, for smoking tobacco, and for other drug use, seen in estimates from epidemiological studies of these behaviors. These estimates often suggest disadvantageous consequences of early onsets and apparent benefits of delayed onsets. Disadvantages are most prominent among US subgroups with early drinking onsets during adolescence, judged by many to be a centrally important developmental stage when adolescents make major progress in brain maturation and toward acquisition of complex social skills (Hall et al., 2016; Schuckit, 2009). In this context, we looked for opportunities to study newly incident underage drinkers and ‘postponers’ who have delayed their 1st drink to age 21, estimating age-specific risks of experiencing a 1st heavy drinking episode soon after drinking onset (i.e., within 12 months after first full drink). We characterize this transition into HED (within 12 months) as a ‘rapid-onset HED’ based on prior studies (Reboussin & Anthony, 2006 Vsevolozhskaya & Anthony, 2015). In addition, recently completed survival analyses disclose a three year median induction interval for HED after drinking onset among newly incident US drinkers (i.e., lag time from age of first full drink to first HED occasion is ≥3 years for ≥50% of drinkers; HG Cheng, CL Wetherington & JC Anthony, 2016, unpublished data).

Our guiding hypothesis included an optimistic expectation of lower HED transition probabilities when adolescent males and females postpone their 1st drink until age 21 years, as compared with larger expected values when corresponding risk estimates are made for underage drinkers. That is, we thought ‘postponers’ might have modulated (dampened) HED transition probabilities in this comparison, in a reflection of (i) assumed benefits of prevailing ‘age 21’ laws, as well as (ii) possibly greater pre-drink vulnerability to maladaptive drinking patterns in subgroups of non-postponing underage drinkers as compared with subgroups of postponers.

As background, we note that almost all prior epidemiological HED estimates address the ‘prevalence’ of being a heavy episodic drinker, with few estimates about the ‘incidence rates’ or risk of experiencing the first HED soon after drinking starts. At present, sharply increasing age-associated gradients in HED prevalence can be seen, with estimated 6% prevalence for 8th-graders, 22% for 12th-graders, and even larger estimates for 20–24 year olds. There also is a general male excess in HED prevalence that might be narrowing (Patrick & Schulenberg, 2010; Patrick et al., 2013).

Based on principles of epidemiological analysis, at population steady state, prevalence proportions for a condition vary as the product of the ‘incidence rate’ for the condition times the mean duration or persistence of the condition once it starts. In consequence, prior HED studies that have estimated prevalence proportions leave us with an inherent ambiguity about incidence processes versus persistence or duration-sustaining processes. In this instance, incidence rates convey results of the process of ‘becoming’ affected (by HED) for the first time, whereas prevalence proportions convey the state of ‘being’ affected (by HED). Estimates from this study are distinctive and differ from prevalence study estimates because our focus is on ‘becoming’ a heavy episodic drinker for the first time relatively soon after the onset of first full drink.

Given this background, our study focuses on age-specific HED transitions observed quite soon after 1st drinks for ‘postponers’ who delay drinking onset to age 21 versus ‘non-postponer’ underage drinkers. Intrigued by previously reported drinking games at 21st birthday parties and other occasions of the 1st drink, we also estimate age-specific risks of very rapid transition to HED such that 1st drink and 1st HED occur in the same month, which might be the birthday month or other celebration of a person’s 1st drink (Rutledge, Park & Sher, 2008). Given the possibility of a narrowing ‘gender gap’ in HED, separate male and female estimates are provided in figures and tables. A ‘epidemiological mutoscope’ view of individual cohort experiences is gained by tracing estimates in the diagonal cells of these tables, as explained elsewhere (Cheng, Cantave & Anthony, 2016a, Cheng, Cantave & Anthony, 2016b; Seedall & Anthony, 2015). Whereas refinement of 21st century ‘epidemiological mutoscope’ approaches has been described previously (Cheng, Cantave & Anthony, 2016a; Seedall & Anthony, 2015), here we note that the mutoscope has its origins in research on tuberculosis death rates conducted by the late Professor Wade Hampton Frost and in studies of cognitive development by Warner Shaie (Frost, 1939; Schaie, 1977).

Methods

Case definitions for heavy episodic drinking

To start our methods section, we wish to bring the reader’s attention to our deliberate and reasoned calibration of our HED case definition, setting a criterion threshold at five drinks per occasion when studying ‘postponers’ versus ‘non-postponers’ and when studying male–female differences. Our reasoning is based on a standard epidemiological research principle that case definitions for incidence rates must first be held constant in order to discover whether there is subgroup variation in incidence rates, with methods-related variations kept to minimum values. This principle applies even though (1) ‘postponers’ to age 21 drinking onsets have more autonomous freedom and a legal right to buy multiple drinks once the 21st birthday anniversary occurs, whereas availability and supply of multiple drinks might be more constrained for ‘non-postponer’ underage drinkers, and (2) in research on adverse health effects of drinking, there is one set of dose (i.e., drink)-response curves for females and a tendency for a rightward-shifted dose–response curve for males (e.g., due to female-male variations in ethanol metabolism, biotransformation, disposition, and excretion). This rightward-shift has motivated use of a 4+ drink threshold for females and a 5+ drink threshold for males in some prior HED research. In this study, if we were to set a ‘4+ drink’ threshold for non-postponers or for females, with a ‘5+ drink’ threshold for postponers or for males, we would distort the initial comparisons, making it necessary to repeat the subgroup analyses with calibration to the same threshold in order to hold constant this methods-related source of variation in the incidence rates. By standardizing the ‘5+ drink’ threshold for all subgroups, these initial estimates speak more clearly to the question of whether postponers to age 21 are more or less likely to make the rapid transition into HED, and to the question of whether males or females are more or less likely to do so. As described below, the data are readily available so that other research teams can pursue this line of research and can set alternative thresholds in future investigations (e.g., 4+ drinks for females versus 5+ drinks for males).

We also make a note about our ‘mutoscope’ approach of tracing the alcohol experience of each cohort down the diagonals of our tables (age by age, and year by year), in a fashion that allows one to check whether the forward progress of a cohort (as shown in the table diagonals) is or is not congruent with the pattern of age-specific estimates (as shown in the table rows). As explained elsewhere, this mutoscope approach assumes a null or negligible ‘period effect’ and is not the same as an age-period-cohort (APC) analysis (Cheng, Cantave & Anthony, 2016a, Cheng, Cantave & Anthony, 2016b). Nonetheless, for interested readers, we provide results from an APC analysis of these data, as explained in the ‘Analysis approach’ section.

Study population, sample, and assessments

The study population is defined to include non-institutionalized community dwelling unit residents aged 12+ years in the US, as studied between 2006 and 2014. Estimates for this study population are from nationally representative samples drawn for the US National Surveys on Drug Use and Health (NSDUH), with annual re-sampling and standardized confidential in-person assessments after IRB-approved parent consent and child assent; detailed methods descriptions are provided elsewhere (Seedall & Anthony, 2015; Substance Abuse and Mental Health Services Administration (SAMHSA), 2012; Substance Abuse and Mental Health Services Administration (SAMHSA), 2015). Participation levels range from 72% to 76%, yielding more than 250,000 12-to-23 year olds. We did not provide estimates for individuals older than 23 years of age because few people in the US start drinking after age 23 years (Cheng, Cantave & Anthony, 2016a, Cheng, Cantave & Anthony, 2016b).

The NSDUH confidential assessments were conducted as audio computer assisted self-interviews (ACASI), each with standardized multi-item modules on health and drugs, including alcohol. Items assessed month, year, and age of 1st full drink and of 1st HED (5+ drinks on a single occasion), and identified >24,000 12-to-23-year-olds who qualified as ‘newly incident drinkers,’ with 1st drinking onset within 12 months before assessment, but with no prior drinking of one full drink before that interval. (NSDUH survey data exist from 2002–2014, but information about the date of the 1st HED is available only since 2006.) The final analytic sample size is 24,100 12-to-23-year-old newly incident drinkers (mean age of drinking onset = 16.5). Table S1 provides a description of unweighted sizes of the samples from which these newly incident drinkers were drawn.

Due to confidentiality concerns, the exact date of interview and birthdates are not available in publicly downloadable NSDUH datasets. Instead, only the quarter of the year when the interview was conducted is available. Therefore, cases of newly incident drinking consist of those who initiated drinking within the past four quarters prior to the assessment, with at least one drink in the 12 months prior to assessment. A similar method was used to identify newly incident HED. Newly incident HED cases had 1st HED within the same 12 months as 1st drink.

Analysis approach

Each age-specific transition probability from 1st consumption of a full drink to 1st HED among newly incident drinkers was conceptualized with an analysis-weighted numerator consisting of newly incident HED cases arising from an analysis-weighted denominator of newly incident drinkers who initiated drinking at a specific age, all events conceptualized to occur within 12 months of the assessment date, including 1st HED occurrences. In this study, we defined rapid transition as the first HED occurring within 12 months of drinking onset, which is consistent with our previous research on drug-related outcomes (Reboussin & Anthony, 2006; Vsevolozhskaya & Anthony, 2015). [We note that NSDUH discloses assessment quarter (but not assessment month). For this reason, the interval for estimation of HED transition probabilities after 1st drink typically is 12 months, but might be as large as 15 months for a small fraction of newly incident drinkers.]

Analogously, for ‘very rapid’ transition from 1st drink to 1st HED, the transition probability was conceptualized with an analysis-weighted numerator consisting of newly incident HED cases whose 1st drink and 1st HED occurred in the same month arising from an analysis-weighted denominator of newly incident drinkers who initiated drinking within 12 months of the assessment date. (In these analyses, the month of 1st full drink is the same as the month of 1st HED, and the 1st HED might have been on the same day as the 1st full drink.)

All estimates are based on NSDUH year-specific analysis weights that account for sample selection probabilities and post-stratification adjustment factors to replicate US Census subpopulation counts. Row-wise, the tabled cells depict age-specific patterns. The mutoscope view is gained by reading the same table cells down the diagonals in evaluation of whether cohort-specific patterns are congruent with age-specific patterns. Standard errors and 95% confidence intervals (CI) are from complex survey delta methods.

Next, logarithms were taken and meta-analysis summaries of age-specific estimates were derived from the nine independent replications. Random-effects estimators were used when heterogeneity was detected (DerSimonian & Laird, 1986; Higgins et al., 2003). As a check on an assumption about null or negligible ‘period effects’ in the interpretation of the mutoscope view, an APC analysis approach known as ‘constrained regression’ was used (Harper, 2015). The constrained regression approach solves an APC problem of model identification via equality constraints. Here, we applied a theoretically plausible (CY2013 = CY2014) period constraint. More details about the use of meta-analysis and APC in the ‘mutoscope’ approach is provided in a previous publication (Cheng, Cantave & Anthony, 2016a).

Results

Figure 1 shows meta-analysis summary estimates for rapid-onset HED transition probabilities within a one-year interval of observation after the 1st full drink, plotted with the age at 1st full drink on the x-axis. For males, one of the estimated peak values in risk of transitioning from 1st drink to 1st HED within that year is seen among ‘postponers’ with age 21 drinking onset (Males age 21: 42%; 95% CI [38–45]). This 42% transition probability estimate is not too distant from expected values based on ‘non-postponer’ underage male drinkers in mid-late adolescence. Lower HED risk estimates are seen for ‘non-postponer’ males with 1st drink before mid-adolescence (i.e., 14 years of age and younger).

Figure 1 Comparison of meta-analytic summary estimates for sex- and age-specific probability (%) of transitioning from 1st drink to 1st heavy episodic drinking among newly incident drinkers who started drinking within 12 months prior to the assessment.

Data from United States National Surveys on Drug Use and Health, 2006–2014. (unweighted n = 23, 735 12-to-21-year-olds). Estimates for 22- and 23-year-olds are presented in Tables 1 and 2, but are not shown due to suboptimal precision associated with the few newly incident drinkers observed in this sample after age 21 years. (A) Any HED in newly incident drinkers. For 16-, 19-, and 20-year-old males, heterogeneity across replications motivated use of the random effects variance estimation approach. (B) Transition to HED within the same month. For 12-, 13-, 18-, 19-, and 20-year-old males as well as 17-year-old females, heterogeneity across replications motivated use of the random effects variance estimation approach.

Among female ‘postponer’ newly incident drinkers who delayed 1st drink to age 21, the estimated risk of experiencing a HED within one year after 1st drink is 27% (95% CI [24–30]). This HED risk estimate is substantially smaller than the comparably observed male estimate at age 21, but is not appreciably different from the peak value among ‘non-postponer’ underage female drinkers in mid-late adolescence (e.g., Fig. 1 Females age 18: 29%; 95% CI [27–32]). Among female newly incident drinkers observed within one year after 1st full drink, HED risk does not vary robustly with age of 1st full drink. A robust male excess in these HED risk estimates can be seen from age 15 onward, but not when underage drinking starts at age 11-to-14-years.

A different pattern is seen when we shift the focus to very rapid transition to HED (i.e., 1st drink and 1st HED occurring in the same month, Fig. 1). In a comparison with underage new drinkers, the risk of this “very rapid” transition to HED is much higher among ‘postponers’ for both males and females (males: 28%, 95% CI [24–31]; females: 18%, 95% CI [15–21]). The corresponding estimates are ∼20% and ∼12% for late-adolescent newly incident male and female drinkers, respectively.

Viewed along the mutoscopic trace down the diagonal cells, Table 1 discloses cohort-specific transitions through adolescence and into the early 20s, based upon re-sampling of each cohort from 2006 through 2014. (Table 2 shows the 95% confidence intervals for the estimated transition probabilities.) In Table 1, for males, as we look down the diagonals from age 16 years in 2006 to age 21 years in 2011, we see the peak HED risk estimate at age 21 years. A similar pattern is seen for other male cohorts transitioning through age 21 years.

Table 1 Estimated sex-, age-, cohort-, and time-specific probability (%, 95% CI) of transitioning from 1st drink to 1st heavy episodic drinking within 12 months of 1st drink.

Data from United States National Surveys on Drug Use and Health, 2006–2014 (unweighted n = 24, 100 12-to-23-year-olds)a.

	Age at First Drinking	
Year	Age = 12	Age = 13	Age = 14	Age = 15	Age = 16	Age = 17	Age = 18	Age = 19	Age = 20	Age = 21	Age = 22	Age = 23	
Panel A. Estimated Transition Probability Among Males (%)	
2006	16 (7,32)	29 (20,40)	35 (25,47)	37 (28,45)	33 (26,40)	38 (30,47)	43 (34,53)	52 (35,68)	61 (42,77)	42 (25,60)	22 (6,55)	37 (4,90)	
2007	5 (2,14)	25 (16,37)	29 (21,37)	41 (35,48)	42 (35,50)	44 (34,54)	41 (31,51)	41 (24,62)	43 (25,63)	35 (23,49)	0 (.,.)	7 (1,42)	
2008	11 (5,25)	35 (24,48)	20 (15,26)	36 (29,44)	52 (42,61)	44 (34,55)	41 (31,52)	40 (26,56)	38 (18,64)	41 (30,54)	12 (2,45)	23 (4,66)	
2009	31 (15,55)	29 (19,42)	26 (18,36)	37 (28,48)	36 (28,45)	47 (36,58)	46 (36,57)	39 (23,57)	24 (15,37)	38 (29,49)	57 (27,83)	22 (3,71)	
2010	20 (10,38)	27 (16,41)	19 (13,27)	36 (28,45)	34 (26,43)	43 (31,56)	41 (32,51)	62 (47,76)	25 (15,39)	39 (29,49)	38 (13,72)	8 (1,48)	
2011	9 (2,32)	12 (6,24)	27 (20,36)	28 (22,36)	35 (28,42)	41 (32,50)	37 (25,51)	30 (18,46)	36 (19,58)	44 (31,57)	27 (9,61)	15 (2,64)	
2012	14 (4,40)	15 (7,28)	25 (17,35)	39 (30,49)	38 (30,47)	41 (29,54)	41 (34,48)	47 (32,63)	40 (22,60)	40 (29,52)	20 (7,46)	20 (2,71)	
2013	27 (11,53)	21 (11,35)	24 (16,33)	33 (23,45)	33 (25,43)	35 (27,44)	35 (26,45)	29 (16,46)	43 (27,61)	37 (25,50)	18 (2,67)	11 (3,36)	
2014	11 (3,35)	14 (6,28)	22 (13,36)	32 (24,41)	48 (39,58)	36 (26,46)	46 (33,60)	55 (32,75)	24 (13,40)	54 (40,66)	16 (2,57)	0 (.,.)	
Meta-analytic summaryb	18(14, 23)	26(22, 30)	25(23, 28)	36(34, 39)	39(35, 44)	41(38, 44)	42(39, 45)	45(38, 53)	37(29, 46)	42 (38, 45)	28(18, 43)	19(13, 30)	
Panel B. Estimated Transition Probability Among Females (%)	
2006	21 (9,40)	23 (14,35)	27 (20,37)	35 (29,43)	27 (20,36)	26 (19,35)	25 (17,34)	29 (15,49)	12 (6,25)	16 (10,27)	22 (6,56)	0 (.,.)	
2007	19 (9,35)	30 (20,43)	29 (22,37)	28 (20,37)	31 (24,39)	35 (23,48)	24 (17,33)	23 (11,42)	17 (8,31)	24 (16,35)	11 (3,38)	35 (9,76)	
2008	21 (9,40)	27 (17,38)	30 (22,40)	26 (20,32)	28 (19,38)	24 (16,33)	29 (21,39)	19 (9,35)	20 (7,45)	26 (16,40)	25 (7,59)	4 (1,28)	
2009	30 (13,54)	26 (18,37)	26 (19,34)	28 (21,36)	27 (19,37)	27 (19,37)	32 (22,42)	34 (20,52)	29 (15,48)	25 (18,35)	0 (.,.)	11 (1,54)	
2010	15 (7,28)	18 (11,28)	28 (21,37)	25 (18,33)	27 (18,38)	34 (24,45)	31 (23,40)	23 (12,41)	28 (17,42)	25 (15,38)	22 (9,46)	50 (15,84)	
2011	29 (14,51)	22 (15,32)	29 (21,38)	31 (23,41)	25 (18,32)	35 (26,45)	31 (24,40)	20 (10,35)	31 (17,51)	33 (24,44)	27 (6,67)	19 (3,67)	
2012	28 (14,47)	24 (10,47)	30 (21,39)	29 (22,38)	28 (21,35)	27 (20,35)	32 (20,46)	23 (12,39)	44 (24,66)	31 (23,40)	19 (7,41)	0 (.,.)	
2013	20 (7,44)	18 (10,29)	25 (18,34)	25 (19,33)	27 (19,37)	26 (19,35)	30 (21,42)	19 (10,33)	20 (10,35)	23 (16,32)	6 (1,32)	0 (.,.)	
2014	13 (4,32)	27 (15,43)	23 (16,33)	26 (18,36)	31 (22,42)	29 (20,40)	30 (23,39)	37 (20,58)	28 (16,45)	31 (21,41)	40 (7,86)	13 (2,56)	
Meta-analytic summaryb	22(18, 28)	24(21, 28)	28(25, 30)	29(26, 31)	28(25, 31)	29(26, 32)	29(27, 32)	26(22, 31)	27(23, 32)	27(24, 30)	23(17, 31)	22(12, 44)	
Notes.

a Cells with the same shade trace the experience of individual cohorts, year by year.

b Meta-analysis summary estimates with each year treated as an independent replication. For 16-, 19-, 20-, and 22-year old males as well as 23-year old females, heterogeneity across replications motivated use of the random effects variance estimation approach.

A generally congruent pattern can be seen via the mutoscope trace of diagonals for newly incident female drinkers, age by age, and year by year. For very rapid transition to HED, the diagonals of Table 2 reveal a highly consistent peak at 21 in all cohorts that traverse age 21 years, for females as well as for males.

To complement the just-reported HED transition probabilities for newly incident drinkers, a set of age-specific unconditional HED population incidence rates for males and females was estimated by re-specifying denominators to include these additional subgroups: (1) all never drinkers at each age, and (2) all past-onset drinkers with no HED experience at each age –as shown in Tables S1–S2. In year-specific and meta-analysis form, these annual unconditional HED population incidence rates are quite low at age 12 years (female: age 12: 0.4%, 95% CI [0.3–1.0]; male: age 12: 0.2%, 95% CI [0.1–0.5]), and show growth to peak values at age 21 years for both females and males (female: 28.7%; 95% CI [27.1–30.3]; male: 33.6%; 95% CI [31.8–35.5]). There is no apparent male–female difference in these unconditional HED population incidence rates prior to age 17 (Fig. S1).

The constrained regression models confirmed our assumption of no tangible ‘period effects’ in either set of estimates. With age and period ‘effects’ adjusted, the estimated ‘cohort effects’ were null. Within the APC framework, with cohort and period ‘effects’ adjusted, the age pattern is congruent with what is described above.

Table 2 Estimated sex-, age-, cohort-, and time-specific probability (%, 95% CI) of transitioning from 1st drink to 1st heavy episodic drinking within the same month of 1st drink.

Data from United States National Surveys on Drug Use and Health, 2006–2014 (unweighted n = 24, 100 12-to-23-year-olds)a.

	Age at First Drinking	
Year	Age = 12	Age = 13	Age = 14	Age = 15	Age = 16	Age = 17	Age = 18	Age = 19	Age = 20	Age = 21	Age = 22	Age = 23	
Panel A. Estimated Transition Probability Among Males (%)	
2006	9 (3,25)	16 (9,26)	16 (8,28)	16 (10,24)	17 (11,25)	22 (16,30)	21 (13,31)	28 (16,46)	33 (18,52)	25 (13,41)	20 (4,56)	0 (.,.)	
2007	2 (<1,15)	13 (6,25)	10 (5,18)	18 (12,25)	21 (14,30)	17 (11,25)	27 (19,38)	27 (13,48)	6 (1,25)	20 (12,31)	0 (.,.)	7 (1,42)	
2008	2 (<1,9)	18 (11,29)	11 (7,17)	12 (8,18)	25 (18,34)	22 (14,31)	28 (19,39)	12 (6,22)	16 (6,38)	34 (24,47)	0 (.,.)	0 (.,.)	
2009	20 (7,44)	18 (9,30)	16 (9,27)	11 (6,18)	17 (11,25)	20 (12,29)	19 (12,30)	10 (4,22)	11 (4,27)	14 (9,22)	44 (9,87)	0 (.,.)	
2010	3 (1,14)	3 (1,11)	10 (5,18)	17 (11,27)	12 (8,20)	19 (12,30)	20 (13,30)	41 (29,53)	7 (2,23)	31 (22,42)	38 (13,72)	8 (1,48)	
2011	0 (.,.)	9 (3,22)	10 (6,18)	13 (9,20)	21 (14,29)	20 (13,29)	16 (9,26)	24 (12,43)	12 (4,31)	30 (19,43)	21 (5,58)	20 (3,69)	
2012	0 (.,.)	1 (<1,7)	13 (7,25)	24 (17,34)	17 (11,24)	21 (12,33)	14 (9,20)	26 (13,46)	10 (5,22)	24 (16,35)	13 (4,36)	20 (2,71)	
2013	12 (3,42)	13 (5,30)	10 (5,18)	15 (10,22)	15 (10,22)	16 (10,24)	11 (6,19)	21 (10,39)	27 (15,45)	29 (18,42)	0 (.,.)	0 (.,.)	
2014	3 (<1,20)	3 (<1,22)	6 (3,15)	18 (10,30)	23 (15,33)	18 (11,28)	19 (10,33)	20 (7,45)	15 (6,31)	33 (22,45)	1 (<1,9)	0 (.,.)	
Meta-analytic summaryb	6 (3, 13)	11 (7, 16)	11 (9, 14)	16 (14, 19)	19 (17, 21)	20 (17, 22)	19 (16, 24)	23 (17, 31)	16 (11, 23)	28 (24, 31)	21 (11, 40)	14 (7, 29)	
Panel B. Estimated Transition Probability Among Females (%)	
2006	15 (5,36)	9 (3,22)	16 (10,26)	16 (11,23)	13 (7,21)	11 (6,19)	9 (5,17)	17 (7,33)	6 (2,17)	9 (4,18)	6 (1,36)	0 (.,.)	
2007	4 (1,10)	6 (3,13)	13 (8,19)	13 (8,19)	13 (8,21)	21 (13,32)	12 (7,20)	12 (5,28)	10 (4,24)	19 (12,30)	11 (3,38)	32 (6,77)	
2008	10 (3,27)	20 (12,33)	9 (6,15)	15 (10,22)	10 (6,17)	5 (3,11)	18 (11,27)	8 (1,32)	6 (1,38)	14 (8,23)	0 (.,.)	6 (1,35)	
2009	6 (1,29)	12 (6,21)	11 (6,17)	13 (8,20)	15 (9,24)	9 (4,16)	18 (12,28)	6 (2,16)	18 (7,38)	20 (12,30)	0 (.,.)	11 (1,54)	
2010	5 (1,17)	7 (3,15)	10 (5,19)	9 (6,13)	14 (7,23)	16 (9,25)	11 (5,20)	5 (2,13)	12 (5,27)	17 (9,29)	15 (8,26)	27 (5,70)	
2011	12 (2,45)	9 (4,18)	10 (6,18)	18 (12,27)	11 (7,17)	20 (11,32)	14 (9,22)	7 (2,20)	16 (6,35)	24 (16,33)	8 (1,34)	0 (.,.)	
2012	16 (5,41)	18 (5,47)	10 (5,21)	11 (7,17)	13 (8,20)	15 (8,25)	14 (8,24)	12 (5,26)	4 (1,12)	18 (12,26)	2 (<1,14)	0 (.,.)	
2013	9 (2,39)	5 (2,12)	6 (3,12)	11 (7,18)	12 (6,20)	12 (7,20)	11 (5,24)	11 (4,23)	9 (3,21)	14 (8,22)	0 (.,.)	0 (.,.)	
2014	3 (<1,18)	15 (7,32)	9 (4,18)	14 (8,22)	19 (11,31)	19 (11,31)	19 (11,31)	15 (8,28)	10 (3,28)	18 (11,29)	40 (7,86)	13 (2,57)	
Meta-analytic summaryb	9 (6, 13)	11 (9, 14)	11 (9, 13)	13 (11, 15)	13 (11, 15)	14 (11, 18)	15 (12, 17)	11 (8, 14)	11 (8, 14)	18 (15, 21)	12 (6, 25)	21 (12, 35)	
Notes.

a Cells with the same shade trace the experience of individual cohorts, year by year.

b Meta-analysis summary estimates with each year treated as an independent replication. For 12-, 13-, 18-, 19-, 20-, and 21-year old males as well as 17- and 22-year-old females, heterogeneity across replications motivated use of the random effects variance estimation approach.

Discussion

What have we discovered? First, studied in aggregate, more than one in three of the US adolescent and young adult newly incident drinkers transition rapidly to HED (20–30% in females and 30–45% in males). Second, for both males and females, the ‘postponers’ who do not drink until age 21 years represent a subgroup of newly incident drinkers who are most likely to engage in heavy episodic drinking during the month of the 1st full drink (Fig. 1B). In addition, ‘postponement’ of 1st drink to the NMDAA-encouraged age 21 threshold does not seem to modulate risk of transitioning into a rapid-onset HED within 12 months after the 1st drink, nor does it provide protection against HED risk, when the comparison group consists of newly incident underage drinkers. These general patterns can be seen in age-specific HED transition probability estimates for both sexes, and also in mutoscopic diagonal traces of each cohort’s maturation through adolescence (Tables 1 and 3). Third, no “gender gap” is observed before mid-adolescence. From age 15 years onward, newly incident male drinkers are more likely than newly incident female drinkers to show rapid onset of 1st HED. This pattern can be seen most clearly in Fig. 1.

As for limitations and counterbalancing strengths, NSDUH relies on self-reports from 12-to-23-year-olds in community survey samples, but is not restricted to school attendees, and therefore should be more generalizable than estimates based on school surveys. In addition, these HED risk estimates lack fine-grained time specifications (e.g., exact dates for birthdate, 1st drink, 1st HED), and do not address a long-term cumulative HED risk as might be seen years after the 1st drink. The NSDUH data are not entirely appropriate for research on cumulative HED incident proportions or HED hazard estimates beyond the first 12–24 months after newly incident drinking. It is possible that the postponers to age 21 experience an unexceptional transition probability during the 1st year after 1st drink, after which the conditional HED risk drops markedly such that the cumulative incidence proportion is not appreciably greater than the corresponding proportion for a more rapid transition into HED. In contrast, the risk of first HED might be distributed over the first 5–9 years of drinking for those who start drinking at age 11–14, such that there is a more gradual accumulation of HED risk, possibly ending up with a cumulative incidence proportion for non-postponer underage drinkers versus postponer ‘age 21’ drinkers. This open question can be resolved via life table analyses to estimate both ‘instantaneous’ hazard of HED after the 1st drink as well as the ‘cumulative’ incidence proportions across multiple years of experience after the 1st drink.

An important counterbalancing strength of this study can be seen in its multiple independent replications to ensure reproducibility. In addition, the large nationally representative samples and use of ACASI boost external and internal validity.

We also note that rapid-onset HED is the only unhealthy drinking consequence studied here. This study’s evidence does not speak to whether delay of the 1st drink to age 21 has other public health benefits, as might be seen in modulation of risk (or prevalence) of other consequences of importance, such as driving under the influence of alcohol (Dejong & Blanchette, 2014; Wechsler & Nelson, 2010).

Notwithstanding considerations such as these, the discovery that 30–40% of newly incident drinkers aged 12-to-23-years-old transition from 1st full drink to 1st HED within roughly one year after 1st full drink might be a noteworty finding for alcohol epidemiology in the US, with implications for future cross-national research in jurisdictions where there is no ‘age 21’ legal minimum. Even though we do not see major age-related differences in HED risk, earlier adolescent drinkers may be particularly vulnerable to cumulative HED risk, as noted above, or to other adverse drinking consequences (Crego et al., 2009; Crews, He & Hodge, 2007; Plunk et al., 2014; Rehm et al., 2010; Spear & Swartzwelder, 2014). The findings throw light on a need for public health actions to prevent HED even when underage drinking has not been prevented, perhaps involving linkages or joint efforts of clinicians, schools, families, and community outreach programs.

This study’s estimates run contrary to our expectation of HED risk modulation via postponement of 1st drink to the NMDAA-encouraged drinking age threshold at 21 years. The resulting evidence deserves consideration in a more thorough discussion of alternatives to the age 21 threshold (Brister, Wetherill & Fromme, 2010; Lewis et al., 2009; Rutledge, Park & Sher, 2008; Wechsler & Nelson, 2010). Similarly, our findings about a very rapid transition to HED stand in stark contrast with our hypothesis about lower risks of HED in ‘postponer’ males and females. Indeed, we found that approximately 30% of ‘postponer’ males and 20% of ‘postponer’ females transition to HED within the same month of drinking onset, the highest among all age groups studied. Although we do not have a solid theory or explanation for this observation, we consider a possibility that previously described 21st birthday drinking games, or non-birthday celebrations of the 1st drink, might play an important role (Rutledge, Park & Sher, 2008). Subject to confirmation and elaboration in future studies, our findings highlight the importance of effective prevention strategies intended to discourage the initiation of drinking games with the first year after drinking onset, including birthday games (Rutledge, Park & Sher, 2008).

In this study, among 12-to-14-year-olds, we found no clear male–female differences in estimates for making transitions into HED within roughly one year after 1st full drink. This finding is in line with previous studies documenting an absence of a ‘gender gap’ in alcohol consumption among early adolescence (Schulte, Ramo & Brown, 2009), including recently published evidence of a female excess risk of starting to drink before the 18th birthday in recent US cohorts (Seedall & Anthony, 2015). This study’s observed male excess in HED from age 15 onward most likely has maturational, social, and biological origins that deserve more study (Keyes, Grant & Hasin, 2008; Kuhn, 2015; Kuntsche et al., 2015; Schwartz, 2013; Seedall & Anthony, 2013). Future studies can inspect physiological changes coincident with the emergence of risk-taking behaviors, as well as learned social roles and expectations, in order to clarify underlying mechanisms for this observed age-specific HED risk variation, which might involve androgens or emergence of gender-related social identities (Byrnes, Miller & Schafer, 1999). In these future studies, facets of the general “gender gap” can be explored usefully by comparing HED estimates based on two approaches: (1) the calibrated approach used here, with the same HED case definition for both sexes, and (2) the sex-differentiated HED metric as can be motivated when the primary research questions involve estimation of toxicity or actual blood alcohol concentration (Wilsnack et al., 2000).

Conclusion

We conclude on a theoretical note with potential clinical practice relevance. Suppose young people who postpone drinking onsets until the 21st birthday truly are at modestly excess HED risk as compared to peers who start consuming the 1st full drink in the teens. Other research teams have commented upon these “21st birthday drinking virgins,” noting that some of them consume too much alcohol as part of the ritual celebration of this transition into adult privileges (Brister, Wetherill & Fromme, 2010; Lewis et al., 2009; Neighbors et al., 2009; Rutledge, Park & Sher, 2008). We theorize that it is not age 21 per se that is crucial. Rather, the important experience might be the rite of passage into drinking on one’s own, whether at age 21 years, or in earlier teen years. To be sure, parents, teachers, and clinicians seeing 20 year olds on the approach to the 21st birthday might try a brief intervention in an attempt to shape the nature of that ritual experience and to dampen its potentially unhealthy facets. Nevertheless, these same brief interventions can be used by parents, teachers, and pediatricians who ask about and discover adolescents who have just started ‘drinking on their own,’ perhaps with deliberate adaptation of the interventions for younger audiences, and with a referral to a ‘smartphone app’ or other electronic technology program after the quick pediatric consult on this topic (Leeman et al., 2015; White et al., 2010). If we are correct, next steps should include more refinement of the increasingly appropriate ‘quick pediatric consult’ approach by medical practitioners or by school counselors and teachers, plus parent-delivered brief intervention approaches. Ultimately, randomized clinical trials of promising refinements in these novel programs will be needed so that their efficacy, effectiveness, and unanticipated externalities can be estimated before widespread dissemination.

Supplemental Information

Supplemental Information 1 Supplementary Material

Click here for additional data file.

The content is the sole responsibility of the authors and does not necessarily represent the official views of MSU, the National Institute on Drug Abuse, or the National Institutes of Health. The authors wish to thank the United States Substance Abuse and Mental Health Services Administration Office of Applied Studies (now the Center for Behavioral Health Statistics and Quality) for completion of its annual nationally representative surveys on drug use and health, as well as its direction and supervision of the annual data gathering and preparation of public use datasets. The authors are grateful for valuable advice and research assistance from Mr. Karl Alcover, as well as helpful suggestions from Dr. Catalina Lopez-Quintero.

Additional Information and Declarations

Competing Interests

Author Contributions

Data Availability

The authors declare there are no competing interests.

Hui G. Cheng conceived and designed the experiments, performed the experiments, analyzed the data, contributed reagents/materials/analysis tools, wrote the paper, prepared figures and/or tables.

James C. Anthony conceived and designed the experiments, wrote the paper, reviewed drafts of the paper.

The following information was supplied regarding data availability:

The raw data can be downloaded from:

https://www.icpsr.umich.edu/icpsrweb/NAHDAP/series/64.

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
