# Peer review of "Does our legal minimum drinking age modulate risk of first heavy drinking episode soon after drinking onset? Epidemiological evidence for the United States, 2006–2014"

_PeerJ, doi:10.7717/peerj.2153_

## Round 0.1 · original submission · Minor Revisions

· Academic Editor

Minor Revisions

Thanks for this interesting paper. Please Addresss the concerns of both reviewers and then resubmit to us.

Reviewer 1 ·

Basic reporting

See comments below.

Experimental design

No comments.

Validity of the findings

See comments below.

Additional comments

The current study highlights an important public health issue relating to risk for heavy episodic drinking among American youth who recently initiated alcohol use. Of note, 20-30% of females and 35-45% of males ages 12-23 years had at least 1 heavy drinking episode (defined as drinking 5 or more drinks in a sitting) within one year of having their first drink. Furthermore, 10-16% of females and 10-28% of males had their first heavy drinking episode within a month of starting to drink. While these findings would be less surprising if the risk for heavy drinking was observed only in individuals who initiated alcohol use at an early age (i.e., had a early age of drinking onset), the authors found that risk also was observed among those individuals who waited until age 21 before having their first drink.

Overall, I think this manuscript makes an important contribution to the literature. However, I have a few questions/comments that I hope the authors might be able to address.

The authors should define heavy episodic drinking the first time it is used in the manuscript.

It is not clear to me why the words prevalence and incidence are in quotations.

Line 86 appears to be missing a word (perhaps “resulting in lower rates” or “producing lower rates”?).

I think the authors can remove their use of underlining and bolding font in the manuscript. The intention appears to be to highlight the importance of the underlined/bolded statements, but I think their importance is clear without the underlining/bolding.

The authors should point out more explicitly why examining risk is different from examining prevalence and what the benefits of examining risk are relative to simply reporting on prevalence rates.

The authors note that they used 5 drinks per occasion as the index of HED for both males and females, stating that “in this epidemiological research, with a focus on female-male differences in the probability of transitioning rapidly from first full drink into HED (as a behavior), we must calibrate the HED criterion and hold it constant.” It is not clear to me why models could not have been run using 4 drinks for women and 5 drinks for men. It seems like comparisons could still be made (as the authors themselves suggest in the discussion). Can the authors explain why this was not a possible and/or desirable analytic strategy for the current study?

Line 140 seems to be missing part of a sentence. “With in-person assessments of after IRB IRB-approved” does not make sense.

The authors bring up the concept of drinking games at 21st birthday parties on several occasions in the manuscript and, in the discussion, state that, “Together with
previously described 21st birthday drinking games, our findings highlight the importance of effective prevention strategies intended to discourage these drinking games (Rutledge et al. 2008).” It is not clear to me exactly how the current findings relate to this topic, due, in part, to the fact that it is not possible to determine what percentage of the sample engaged in a heavy drinking episode the very first time they drank. Furthermore, while playing drinking games makes it more likely that an individual would engage in a heavy drinking episode, it is not clear to me why the focus is on drinking games at 21st birthday parties specifically. Are the authors proposing this is why individuals who wait until they are 21 experience risk for HED? It does not seem that the authors actually have any data to speak to this issue based on their sample.

A general note, some of the sentences in the manuscript read in a choppy manner (e.g. “Participation level ranges were 72%-76%, yielding over 250,000 12-to-23 year olds, the
age range under study here because few start drinking after age 23 years.). The authors should carefully read the manuscript to identify areas where sentence structure/flow can be improved. Also, some of the language in the manuscript feels oddly informal for a scientific paper (e.g., What have we discovered?).

Reviewer 2 ·

Basic reporting

Helpful overview of US alcohol legislation in addition to previous estimates of HED prevalence provide a informative introduction.

May be helpful to expand on the neurobiological vulnerabilities with the initiation and continued use of alcohol posed to this population group.

Could include mean age of sample.

Table 1 could include the CI from table 2 to reduce number of tables.

Experimental design

Could provide more of a justification for why a 12 month period from alcohol initiation to 1st HED is considered to be 'rapid?

Knowledge gap being investigated is highlighted.

Validity of the findings

Clear discussion section which highlights the key findings of the study

There are clear discussion points made with regards to the state-level 'age 21' drinking laws and that this threshold was not found to modulate the risk of rapid transition to HED which is further expanded on in the conclusion. The title of the journal may wish to reflect more around this 'age 21' being a focus of this paper.

Additional comments

Unfamiliar with the term 'mutoscope'?

---

## Round 0.2 · accepted · Accept

· Academic Editor

Accept

Thank you for your quick responses to the comments of our reviewers. We are satisfied that you have addressed all of the concerns raised and therefore are pleased to accept your paper for publication.